# The Pros and Cons of Using Oat in a Gluten-Free Diet for Celiac Patients

**DOI:** 10.3390/nu11102345

**Published:** 2019-10-02

**Authors:** Iva Hoffmanová, Daniel Sánchez, Adéla Szczepanková, Helena Tlaskalová-Hogenová

**Affiliations:** 12nd Department of Internal Medicine, University Hospital Královské Vinohrady and Third Faculty of Medicine, Charles University, Ruská 87, 10000 Prague, Czech Republic; iva.hoffmanova@fnkv.cz; 2Laboratory of Cellular and Molecular Immunology, Institute of Microbiology of the Czech Academy of Sciences, Vídeňská 1083, 14220 Prague, Czech Republic; szczepankova.adela@gmail.com (A.S.); tlaskalo@biomed.cas.cz (H.T.-H.); 3First Faculty of Medicine, Charles University, Kateřinská 1660/32, 121 08 Prague, Czech Republic

**Keywords:** amylase/trypsin inhibitors, celiac disease, gluten-free diet, gluten-free oat, oat

## Abstract

A therapeutic gluten-free diet often has nutritional limitations. Nutritional qualities such as high protein content, the presence of biologically active and beneficial substances (fiber, beta-glucans, polyunsaturated fatty acids, essential amino acids, antioxidants, vitamins, and minerals), and tolerance by the majority of celiac patients make oat popular for use in gluten-free diet. The health risk of long-time consumption of oat by celiac patients is a matter of debate. The introduction of oat into the diet is only recommended for celiac patients in remission. Furthermore, not every variety of oat is also appropriate for a gluten-free diet. The risk of sensitization and an adverse immunologically mediated reaction is a real threat in some celiac patients. Several unsolved issues still exist which include the following: (1) determination of the susceptibility markers for the subgroup of celiac patients who are at risk because they do not tolerate dietary oat, (2) identification of suitable varieties of oat and estimating the safe dose of oat for the diet, and (3) optimization of methods for detecting the gliadin contamination in raw oat used in a gluten-free diet.

## 1. Introduction

Current gluten-free diets often have nutritional limitations. The nutritional and health benefit of oat is associated with an increased intake of dietary fiber, water-soluble beta glucans, a positive ratio between saturated and unsaturated fatty acids, polyunsaturated fatty acids, essential amino acids, antioxidants, vitamins, and minerals. These benefits promote the use of oat in a gluten-free diet for celiac patients [1]. Moreover, a diet enriched with oat is also tastier in contrast to commercially available products commonly used in gluten-free diets. The health, nutritional, and technological advantages of oat has led to the acceptance of oat as foodstuff suitable for a gluten-free diet in Europe (2009, COMMISSION REGULATION (EC), No. 41/2009) and the USA (2013, FDA regulation) [2].

Although oat is a regular component of a gluten-free diet for celiac patients in remission, there are still many scientific and clinical questions to be answered.

This paper attempts to contribute to the discussion about the perspective of oat in a gluten-free diet. We try to integrate nutritional, immunogenic, and clinical aspect of dietary oat for celiac patients in remission, as well as the pitfalls of food processing, oat contamination, and the issues of detection of toxic prolamins contamination in dietary oats. Moreover, we point out the potential role of oat amylase/trypsin inhibitors in incomplete clinical remission of celiac disease, and in persistent minor mucosal injury. Oat amylase/trypsin inhibitors have not yet been sufficiently studied and discussed in the literature in this context. In addition, we emphasis a risk of hypersensitivity to molecules of some oat varieties in individual celiac patients. Finally, we try to define further directions necessary to better clinical evaluation of the safety of oat in a gluten-free diet. 

## 2. Beneficial Effects of Oat in a Gluten-Free Diet

Wheat is responsible for the induction of several diseases, such as celiac disease, wheat allergy, and non-celiac gluten sensitivity [3,4,5,6,7]. The prevalence of these gluten-related diseases has reached 1% to 6% in Europe and North America [8,9,10,11,12,13]. Celiac disease is induced in genetically susceptible individuals by ingestion of the wheat gluten, including alcohol-soluble (gliadin) and alcohol-insoluble fraction, and phylogenetically-related cereal prolamins (mainly hordeins in barley and secalins in rye) [14,15]. The alimentary intake of gluten induces, in celiac disease patients, intestinal mucosa damage characterized by intraepithelial lymphocytosis, villous atrophy, and crypt hyperplasia in the duodenum and jejunum and, subsequently, the loss of digestive and barrier functions accompanied by various gastrointestinal and extraintestinal symptoms. A lifelong gluten-free diet is the only effective treatment for celiac disease. Adherence to a gluten-free diet leads to recovery from mucosal damage and the disappearance of the serological markers of celiac disease, i.e., antibodies against tissue transglutaminase, endomysium, and deamidated gliadin peptides [14,15,16,17,18,19].

Oat (*Avena sativa* L.) caryopsis possesses one-third more protein (15% to 20%) and four times more lipid (5% to 9%) than conventional cereals such as wheat, rye, and barley. The oat caryopsis also contains relatively high amounts of fiber (12% to 14%) and beta glucans (5%) [20,21]. Moreover, oat kernels are rich in unsaturated fatty acids, polyunsaturated fatty acids, essential amino acids, antioxidants, vitamins, minerals (iron, potassium, and calcium), and low molecular weight soluble phenolic avenanthramides; these components of oat are not present in other cereals [22,23]. The individual cultivars of oat possess varying amounts of biologically active compounds, for example, the oat cultivar, Golozrni, is rich in total phenols and flavonoids, possesses exceptional antioxidant capabilities, has considerable reducing power, and antihyperglycemic activity [24]. It has been shown that consumption of oat grain or oat bran can decrease total plasma cholesterol and LDL-cholesterol levels, suppress inflammation, relaxes arteries, reduce atherosclerosis, and reduce colon cancer risk in some individuals [25,26,27,28].

Despite a large amount of proteins, the proportion of avenins (prolamins) in total oat grain proteins is less (10% to 15%) than that of gliadins in wheat (80% to 85%), secalins in rye, and hordeins in barley. The proline content in avenins is also less than that found in gliadins, secalins, and hordeins [29,30]. Avenins are also more easily digested by gastrointestinal proteases, which is in contrast to gliadins, secalins, and hordeins. Moreover, the peptides arising from protease digestion of oat protein has been shown to have less affinity for MHC II gp coded by HLA-DQ2.5 haplotype, which is associated with celiac disease. These properties of avenins clearly reduce their immunogenicity and toxicity for celiac patients as compared with wheat, rye, and barley prolamins [30,31]. Recently, several studies documented the suitability of oat in a gluten-free diet for celiac patients. Pinto-Sánchez et al. (2017) [2] and Lionetti et al. (2018) [32] postulated that there was no risk associated with the consumption of oat products by both adult and pediatric celiac patients in remission (i.e., patients who had been adhering to a gluten-free diet for an extended period of time). These authors did not find significant pathological changes in intestinal histology and in immune parameters (e.g., seronegativity for antibodies against tissue transglutaminase) in celiac patients consuming oat. Additionally, they found no symptomatic manifestation in remitted celiac patients after consuming oat [2,29,32].

## 3. Individual Hypersensitivity to Oat in Celiac Patients

Nevertheless, the consumption of oat is recommended only for celiac patients in remission and even then, cautiously. Immunogenicity (toxicity) of certain varieties of oat for celiac patients has been discussed, and adverse immune reactions in some celiac patients against oat proteins can occur [30]. A study by Tuire et al. (2012) documented an association between the consumption of oat and health problems in some celiac patients in remission. Histological analysis of duodenal and jejunal biopsies in 96 out of 170 celiac patients on a gluten-free diet that included oat showed persistent intraepithelial lymphocytosis [33]. Fifty grams per day of dietary oat over 12 weeks caused in one out of 19 celiac patients on a gluten-free diet partial villous atrophy of the jejunal mucosa as well as skin exanthema [34]. The selectivity and specificity of the immune response against oat proteins in patients with celiac disease were documented by the presence of avenin-specific T-cells in the intestinal mucosa of five celiac patients sensitive to oat proteins [35]. Six stimulatory (immunogenic) sequences of avenins for T-cells were identified in celiac patients exposed to three days of dietary oat [18]. The most immunogenic oat avenins, for T-cells in celiac patients, were the gamma-3 and gamma-4 avenins containing the QQQP, QQQQ, and PSQQ motifs (P, proline; Q, glutamine; and S, serine), however, only eight percent of celiac patients possess the T-cells specific for avenins in celiac patients sensitive to oat and long-term consumption of oat (100 g per day) by celiac patients elicited only weak activation of the T-cells specific for avenins [31]. Moreover, it has been suggested that substantial differences in the degree of avenin-induced activation of T-lymphocytes exist between oat varieties [36,37,38,39]. Avenin-poor oat varieties may also be less immunogenic for celiac patients as compared with avenin-rich varieties [18,36,37,38,39,40]. Interestingly, cross-reactivity of T-cells specific for gliadin with avenins after an oat challenge (exposure) in celiac patients in remission was not found [29,37]. Nevertheless, rare adverse immune reaction in celiac patients exposed to dietary oat can occur and, for this reason, the introduction of oat into the diet of celiac patients should be done with care [1]. On the basis of recent evidence, celiac patients should only start to consume oat after establishing that they are in clinical, serological, and biochemical remission [2,33,41]. The absence of sideropenic anemia, hypovitaminosis, secondary hyperparathyroidism, autoantibodies against tissue transglutaminase, and deamidated gliadin peptides is necessary before the introduction of oat into a gluten-free diet. An adverse immune reaction associated with the consumption of oat or elevation in anti-tissue transglutaminase autoantibodies indicate that oat needs to be excluded from the diet [1,37,38,39].

## 4. Contamination of Oat by Cereal Prolamins

The contamination of oat by wheat, barley, or rye is probably the main limitation to its use for a gluten-free diet. Indeed, gluten contamination of oat occurs frequently. Typically, commercially available oats are not suitable in a gluten-free diet for celiac patients due to their routine contamination with wheat, rye, or barley. Only gluten-free oat is acceptable as a foodstuff for celiac patients. The cultivation and processing of gluten-free oat requires sophisticated technology. The prevention of contamination of oat by wheat, rye, or barley includes topographically separate fields with a suitable distance and a natural barrier between the fields sown with these gluten-containing plants, and separate harvesting and spatially isolated technological processing of oat from wheat, rye, and barley grains. Special attention must be paid to the purity of oat seeds, as well as agricultural and food industry facilities, which must be oat specific. Moreover, a field previously planted with wheat, rye, or barley cannot be used for oat for at least eight years. Oat fields must also be routinely inspected for contaminating cereal plants (containing prolamins immunogenic for celiac patients), and those plants have to be removed [1,38]. The AOECS (Association of European Coeliac Societies) is entitled to guarantee gluten-free product derivatives prepared from oat in Europe. Gluten-free oat must meet the legislative criteria for gluten-free foodstuff, i.e., the content of gluten in the end-products must be less than 20 mg/kg [42].

The suitability of foodstuffs for a gluten-free diet is traditionally estimated on the basis of a gliadin-specific ELISA test, mainly ELISA R5 or ELISA G12. However, several factors could complicate the objectivity of these test results despite their high specificity and sensitivity. The antibodies in the immunological tests do not exhibit equal affinity to toxic prolamins from wheat, rye, and barley. Antibodies against the repetitive amino acid sequence of proline and glutamine of toxic wheat prolamins can exhibit immunoreactivity to prolamins from wheat, rye, barley, and oat [43]. Nevertheless, ELISA R5 shows virtually no cross-reactivity to oats and can, therefore, be used to assess wheat, rye, or barley contamination in oats, whereas ELISA G12 is known to cross-react to certain oat cultivars, and the results of the ELISA have been shown to differ in various technologically processed food, for example, soy-based sauces showed nonspecific inhibition with the R5 and G12 antibodies. Moreover, differences exist in recognizing various categories of fermented and hydrolyzed foods in the test using R5 and G12 antibodies, as well as antibodies 2D4, MIoBS, and Skerritt [44]. The ELISA responses showed high variability depending on the type of cereal proteins and the antibody used; ω1,2-gliadins and γ-75k-secalins were most reactive, whereas ω5-gliadins and γ-, B- and D-hordeins were detected with the lowest sensitivities [45]. The gluten contents quantified by gel-permeation high-performance liquid chromatography with fluorescence detection as the sum of gliadins and glutenins were higher than those revealed by R5 ELISA in 19 out of 26 wheat starch [46]. Recently, a sensitive immunochromatographic test for the detection of gluten (RIDA^®^QUICK Gliadin) was successfully tested for detection of gliadin in foods, on surfaces, and in cleaning in place waters [47]. Several types of immunosensors have been developed by Funari et al. [48], Manfredi et al. [49], and Ng et al. [50]. Proteomic methods offer alternative approaches for quantification of toxic prolamin proteins. In addition, Schalk et al. described a stable isotope dilution assay followed by liquid chromatography tandem mass spectrometry for detection of 33-mer gluten peptide in wheat species and cultivars [51]. Vatansever et al. (2017) developed a new, fast and robust micro in pair LC-MS analytical method in combination with denaturation agent RapidGest™ (prior to the enzymatic digestion) for the qualitative and quantitative determination of 30-mer toxic gliadin peptides in wheat flour [52]. The use of these proteomic methods, however, requires comprehensive and well annotated sequence databases for gluten. A manually curated database of gluten proteins (GluPro V1.0) in a FASTA format has been performed by Bromilow et al. supporting the development and reliability of proteomic methods detecting gluten proteins [53]. Concerning DNA analysis, two different DNA amplification techniques, real-time PCR (qPCR) and real-time loop-mediated isothermal amplification (qLAMP) for detection and quantification of gluten were presented in the study by Garrido-Maestu et al. [54] (2018).

Several monoclonal antibodies, both mouse (IgG1 and IgM) and humanized hybrid (IgA1), against gliadin and its peptides were prepared and analyzed for their fine specificity using the Pepscan method with overlapping decapeptides of alpha-gliadin, gamma-gliadin, and omega-secalin. Moreover, several of the peptides carried Q–E substitution at defined position. Our experimental works document that the target of tested anti-gliadin antibodies were as follows: mouse monoclonal antibodies R5, R1, R4, 6H5, 8D4, 1C6, 5B10, 4D6, 7C6, 5C7, and 6H5; and humanized mouse monoclonal IgA antibodies 8D12, 15B9, 9D12, and 18E2 recognized repetitive sequences containing the motif QPFPXQ (X = Q, L, P); but also peptides containing the sequences QQSFPQQ, QQTFPQP, and QPFRPQ. A comparison of immunodominant antigenic sites among the IgA humanized anti-gliadin antibodies, and the reactivity of IgA from active celiac patients revealed a common immunodominant region on alpha-gliadin localized to the N-terminal part with the motifs QFQGQQQPFPPQQPYPQPQPFP and QPFPSQQPYLQL. Nevertheless, the majority of the humanized IgA antibodies recognized the sequences localized in the most significant pathogenic sequence of alpha-gliadin, i.e., 33-mer: LQLQPFPQPQ and PQLPYPQPQPFL, which is localized in position p31–43. Only R5 anti-gliadin antibodies are widely used in commercially available tests. The reactivity of tested monoclonal anti-gliadin antibodies showed similar binding properties to gliadin and secalin, probably driven by recognition of the peptide core sequences QQQ and PFP, which represents the putative immunodominant structure in cereal prolamins. Nevertheless, the main epitope recognized by R5 was QQPFP. The effects of substitution of Q by E on binding of R5 were investigated. Introduction of E residue directly in front of the PFP motif decreased antibody binding strongly [55,56,57]. The database of the U.S. National Center for Biotechnology Information was searched for exact matches with the QQPFP motif. Repetitions of this sequence motif predominantly occur in Triticeae (Triticum, Hordeum, Secale, Aegilops). The sole motif was found in Oryza sativa and Sorghum bicolor, whereas the sequence QQPFP was not found in Avena and maize [55].

## 5. The Possible Similarity between Oat and Wheat Amylase/Trypsin Inhibitors

Patients with celiac disease develop a gluten-specific immune response; nonetheless non-gluten proteins also contribute to the pathophysiology of wheat-related disorders [3,18]. Seeds of several cereal species, such as wheat, barley, rye, triticale, and oat contain proteinaceous alpha amylase/trypsin inhibitors, which are pest resistance molecules. The alpha-amylase/trypsin inhibitors contain from 120 to 160 amino acid residues, which occur as monomeric, homodimeric, or heterotetrameric molecules differentially active against alpha-amylases from *Coleoptera* and *Lepidoptera* phytophagous species [58,59,60]. Thirty-seven oat alpha-amylase/trypsin inhibitor species, with molecular weight about 14 kDa, have been classified into 3 groups: alpha-amylase/trypsin-inhibitors 1, alpha-amylase/trypsin-inhibitors 2, and alpha-amylase/trypsin-inhibitors 3. Oat alpha-amylase/trypsin inhibitors 1, 2, and 3 (of approximately 30% identity) have different primary structures, molecular weights and, isoelectric points (pI). These inhibitors exhibit the highest sequence similarity with the components of the wheat tetrameric amylase/trypsin inhibitors, which are referred to as CM (chloroform/ethanol mixture solubility) proteins [61,62,63]. Each wheat amylase/trypsin inhibitors, about 60 kDa in size, contains one copy of either the CM1 or CM2 protein, one copy of either the CM16 or CM17 protein, and two copies of the CM3 protein. Concerning the sequence comparison, oat alpha-amylase/trypsin inhibitor 1 possesses the highest similarity (56%) with the wheat CM16 inhibitor, whereas oat alpha-amylase/trypsin inhibitor 2 shows approximately 60% similarity with wheat CM1, CM2, and CM16 amylase/trypsin inhibitors. By contrast, a similarity of approximately 25% has been found between oat alpha-amylase/trypsin inhibitors and wheat alpha-amylase inhibitor 0.28 and alpha-amylase inhibitor 0.19 [49,50,52]. The amount of the alpha-amylase/trypsin inhibitors is different during the development of caryopses in the vegetative period. The maximum inhibitor amount in the starch granules of caryopses is reached 35 days after anthesis [61,62].

### Cereal Amylase/Trypsin Inhibitors in Celiac Disease and Potential Role in Incomplete Remission of Celiac Patients on a Gluten-Free Diet

Interestingly, wheat amylase/trypsin inhibitor CM3 and alpha-amylase inhibitors 0.19 and 0.28 have been recently identified as potent activators of the innate immune response. These inhibitors have stimulated human dendritic cells, derived from the monocytes of peripheral blood from both patients with active celiac disease and those on a gluten-free diet, to secrete IL-8. Consistently, enterobiopsy specimens from celiac patients in remission (i.e., those adhering to a gluten-free diet for more than 6 months) cultivated in a medium with wheat alpha-amylase/trypsin inhibitors have shown an increased expression of IL-8 mRNA. Moreover, these inhibitors have stimulated also monocyte-derived dendritic cells from healthy controls to produce IL-12. The adjuvant effect of wheat alpha-amylase/trypsin inhibitors (including alpha-amylase inhibitors 0.19 and 0.28) on human innate immune cells has been mediated by their interaction with the TLR4-MD2-CD14 complex [64,65,66]. Nevertheless, deproteinized wheat starch is used for gluten-free food production, without taking into account the possible source of alpha-amylase/trypsin inhibitors, which are potentially pathogenic in celiac disease and wheat allergy. 

In addition to innate immune cells, wheat alpha-amylase inhibitors 0.19 and 0.28 have been also shown to stimulate adaptive immune cells, i.e., B-cells in celiac patients. B-cells combine the innate and adaptive immune response, reacting sensitively to changes in antigenic content throughout life. The presence of serum IgA, IgG, and IgE antibodies against alpha-amylase inhibitor 0.19 and 0.28 has been recently described in patients with active celiac disease, including in some of those patients on a gluten-free diet [67]. The persistence of seropositivity to wheat alpha-amylase inhibitors 0.19 and 0.28 in celiac patients on a gluten-free diet can be speculatively explained as a consequence of contamination of non-gluten proteins in gluten-free foodstuff, e.g., in deproteinized wheat starch [67].

Wheat gliadins are not the only molecules recognized by the serum antibodies of celiac patients. Huebener et al. (2015) found serum IgA and IgG antibodies, in celiac disease patients, that recognized a number of non-gluten molecules extracted from U.S. hard red spring wheat *Triticum aestivum* Butte 86 flour. These include serpins, purinins, globulins, farinins, and several amylase/protease inhibitors [68]. The B-cells respond sensitively and dynamically to antigenic stimuli throughout the human life by changing the affinity, specificity, and isotype of produced antibodies, which may change their biological activity (i.e., pathogenic vs. protective). If we accept the long-term presence of alpha-amylase inhibitors (including those in oat) in celiac patients, there exists the potential for continuous stimulation of the immune system of celiac patients on a gluten-free diet, which could lead to subclinical inflammation and autoreactivity. Histological recovery is usually long-lasting, and it can be incomplete in some patients with celiac disease on a gluten-free diet, which reflects the persistence of celiac disease symptoms and seropositivity for antibodies against tissue transglutaminase in these patients [69,70,71,72,73]. Interestingly, in a study with adult patients with non-celiac gluten sensitivity, 24% had uncontrolled symptoms despite consuming a gluten-free diet [12,74].

It is assumed that the high affinity (avidity) complexes of antigen B-cell receptor (i.e., surface antibodies associated with signaling molecules) could promote an extrafollicular B-cell response leading to an increasing number of plasma cells secreting antigen-specific antibodies [75]. Interestingly, an extrafollicular response or a short-time germinal response is assumed in the development of specific B-cells producing the low avidity autoantibodies against the key autoantigen of celiac disease, i.e., tissue transglutaminase [76,77]. Unfortunately, the histological analysis of the small intestinal mucosa is not usually performed in a follow-up of these patients. In mouse models, wheat alpha-amylase/trypsin inhibitors induced the innate immune response by activation of the TLR4-MD2-CD14 complex that led to barrier dysfunction and mucosal damage. The addition of wheat alpha-amylase/trypsin inhibitors to the diet of gluten-sensitized mice expressing HLA-DQ8 increased intestinal inflammation. Interestingly, the pathogenic effect of wheat alpha-amylase/trypsin inhibitors in this animal model was mitigated by *Lactobacilli*, whose proteases probably degraded the inhibitors [78].

Nevertheless, the influence of oat alpha-amylase/trypsin inhibitors on innate immune bioactivity, i.e., the capacity to release IL-8, CCL2 (MCP-1), and TNF-alpha from THP-1 cells was also analyzed. Wheat (*Triticum* ssp.), barley (*Hordeum vulgare* L.), and rye (*Secale cereale*) effectively stimulated THP-1 cells to secret Il-8, CCL2 (MCP-1), and TNF-alpha. Interestingly, the inhibitors of soya (*Glycine max*), millet (*Pennisetum glaucum*), and teff (*Eragrostis tef*) were five times less active in stimulating the secretion of these cytokines than the inhibitors of lentils (*Lens culinaris*), quinoa (*Chenopodium quinoa* Willd.), and oat (*Avena sativa*), as much as ten times less active than the inhibitors of amaranth (*Amaranthus* L.), rice (*Oryza sativa*), and corn (*Zea mays*), and twenty times less active than wheat, barley, and rye. Simultaneously, the author assumed none or very little capacity for avenins to induce an adaptive immune response. Nevertheless, the alpha-amylase/trypsin inhibitors isolated from various wheat cultivars showed different bioactivity, i.e., capacity to induce the release of IL-8, CCL2 (MCP-1), and TNF-alpha from THP-1 cells. Structurally, alpha-amylase/trypsin inhibitors are different: Kunitz-type includes wheat (*Triticum* ssp.), barley (*Hordeum Vulgare* L.), rye (*Secale cereale*), rice (*Oryza sativa*); Lectin-type includes soya (*Glycine max*), lentils (*Lens culinaris*); Knottin-type includes amaranth (*Amaranthus* L.); and Thaumatin-typ includes corn (*Zea mays*). Structurally, alpha-amylase/trypsin inhibitors of oat (*Avena sativa*), buckwheat (*Fagopyru esculentum*), and quinoa (*Chenopodium quinoa*) are still unclassified [66].

## 6. Factors Influencing the Safety of Oat for Human Alimentation

Several studies documented that pure oat is safe for most celiac patients [37,39,40], however, several factors may influence immune sensitization of humans to oat proteins towards allergy or other immune-mediated intolerance. The key biological mechanisms mediating this risk are changes of the transcription profile leading to enrichment of (modified) proteins atypical for the mature grain. Salt stress of oat plants may lead to the expression of allergens or proteins capable of triggering adverse reactions (intolerance) [79]. An example is an oat variety that is highly resistant to herbicides and pests. These cultivars can abundantly express a large number of multifunctional proteins with enzymatic activity, which can provoke oat allergy and intolerance [80]. The proteomic profile of oat seeds toward the expression of potentially immunoactive substances can be substantially changed by storage conditions such as moisture and temperature [81,82]. Finally, the food industry makes use of many artificial chemical substances, which may have immunomodulatory properties. Acetylated oat starch, as well as deamidated and succinylated oat proteins are found in 20% of oat cakes [83]. 

## 7. Conclusions

The nutritional value of oat predisposes its use in gluten-free diet for celiac patients, however, the safety of oat in a gluten-free diet is still a matter of some contention. Gluten-free oat is a beneficial component of a diet only for celiac patients in clinical, serological, and histological remission. The safe dose of dietary oat for celiac patients in remission ranges from 20 g (for children) to 70 g (for adults) per day. The introduction of oat into a gluten-free diet for newly diagnosed celiac patients is not appropriate since a strict adherence to a gluten-free and oat-free diet is required for newly diagnosed celiac patients [37,38,39]. Celiac patients in remission receiving food supplemented with oat as part of a gluten-free diet should do so under medical supervision due to individual susceptibility to oat [1,37,38,39]. Although the dietary oat (without contamination with gliadins) is tolerated by the majority of celiac patients, the individual sensitivity to oat cannot be excluded. Testing of the potential sensitivity to dietary oat in celiac patients requires the development of suitable methodological approaches that can estimate the risk of individual oat varieties in celiac patients. These approaches should be targeted on the following: (1) immunological reactivity of patients against the proteome components of oat (e.g., the testing of preformed specific antibodies, at least of the IgE isotype), (2) physicochemical properties of oat proteins (e.g., protease digestibility, pH degradability, and biological half-life), and (3) biological activity of the proteins and glycans (enzymatic properties and adjuvant influence on innate immune cells, e.g., amylase/protease inhibitors). While all these are important, the main problem lies in the agriculture and food industry, which must produce both gluten-free oat and oat foodstuffs. Moreover, the development of more sensitive and specific tests for the detection of gluten (wheat) contamination in oat products is needed.

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
