# Peer review of "The Pros and Cons of Using Oat in a Gluten-Free Diet for Celiac Patients"

_nutrients, 2019, doi:10.3390/nu11102345_

Round 1

Reviewer 1 Report

This review summarizes the pros and cons of using oat in the gluten-free diet for celiac disease patients. The paper is written reasonably well, but it contains quite some speculation in point 4 on oat ATIs. From the literature, there is very little (if any at all) information on the role of oat ATIs in possible immunoreactivity. Therefore, this part seems to be quite long, although there is only little clear evidence.

It would have been nice to have a table and/or a figure in this work, instead of only text.

Comments

L44: Not only the alcohol-soluble fraction is known to induce celiac disease, but also the alcohol-insoluble fraction known as glutelins; please add and correct

L131-135: This part has some weaknesses. The ELISA R5 has virtually no cross-reactivity to oats and can, therefore, be used to assess wheat, rye or barley contamination in oats. However, the ELISA G12 is known to cross-react to certain oat cultivars and for this reason the test is unsuitable for the same task. There are also some novel developments, both in analytical ELISA methods, as well as in non-ELISA methods. These approaches should be included and discussed more, especially because the contamination issue is very serious in oats and might even be responsible for the perpetual question of whether oats are safe or not

L148ff: Here, you should clearly differentiate between your antibodies and the commercially available antibodies.

L149: Only three amino acids seems very short for a core sequence. Are you sure? How was this determined and which antibody does this refer to? The R5 epitope is QQPFP.

L154: This part should be checked very carefully. NIRS can be used to determine flour protein content, but certainly cannot be used to detect trace levels of gluten in foods. Then, proteins and peptides are not volatile and thus totally unsuitable for gas chromatography. The method to detect gluten (peptides) is liquid chromatography mass spectrometry. Please correct and add appropriate literature references.

L158ff: This whole paragraph on oat ATIs is rather strange, because it is totally unclear what you are trying to imply here. So far, there is little to no evidence that oat ATIs might cause an immune reaction.

L165: In wheat, ATIs are also found in significant amounts within gluten.

L186ff: Here, it would be helpful to also have % identities.

L191ff: Which sequence similarity would you consider relevant in this case? In wheat proteins, you might have a sequence similarity of 60-70%, but still a completely different protein with different immunoreactive properties. So here, again, it is unclear what you are trying to imply.

L217ff: This part reads more like an original research article, but this is out of place in a review/discussion paper.

L232: What do you mean with “changing their role”?

L282: Here, again, it is not clear what you are trying to say. Is this treatment per se harmful? I’d rather think not.

Funding: Funding sources need to be declared, or else the statement made that this paper received no funding. Probably the projects mentioned in the acknowledgements should be here.

Author Response

Dear Reviewer 1,

We are grateful for the comprehensive and important comments. We have corrected the text after recommendation and requirements of you. We revised the manuscript carefully point-by-point, accepted and tried to meet all the recommendations and orders/requirements of the reviewer. All the changes are clearly indicated (in yellow) in the revised manuscript.

L44: We corrected the information after recommendation of the reviewer

L131-135: We corrected the information concerning ELISA R5 and ELISA R12 and included information about “novel developments, both in analytical ELISA methods, as well as in non-ELISA methods” after requirements of the reviewer. Simultaneously, we corrected the information contained in paragraph: L154

L148ff: We modified this paragraph and introduced the antibodies which we used in our studies.

L149: We completed the information concerning analysis of specificity of anti-gliadin antibodies.

L158ff + L165: We significantly reduced information about ATIs.

L186: We add the information about percentage of identities about oat ATIs.

L191ff: In this paragraph, we wanted only to place the information about sequential similarities among potentially pathogenetic cereal non-gluten proteins in a broader biological context. In this paragraph we only state the existence of certain similarities among wheat amylase/trypsin inhibitors potentially recently associated with pathogenesis of celiac disease.

L217ff: No experimental data has not been included in the text. For this reason, we reformulated this paragraph.

L232: We agree with the reviewer that our expression in this sentence is confusing. We corrected the sentence. “The B-cells respond sensitively and dynamically to antigenic stimuli throughout the human life by changing the affinity, specificity, and isotype of produced antibodies, which may change their biological activity (i.e. pathogenic vs. protective).”

L282: We agree with the reviewer that the sentence is deceptive. We corrected the sentence towards: “Moreover, also storage conditions such as moisture and temperature substantially change the proteomic profile of oat seeds toward the expression of potentially immunoactive substances [81,82].”

We thank the reviewer for his notice concerning Funding sources. We have already complemented the subsidizing sources of this manuscript.

Reviewer 2 Report

The paper is an interesting analysis of oat for celiac patients.

Several points need to be addressed and clarified.

You specify the paper is a discussion. Is this paper type foreseen by the Journal? If by "discussion" you mean a sort of critical review, why are you presenting some outcomes of An experimental study you claimed carrying out (see line 136 to 157 ).  The introduction needs reformulation, so that you introduce the reader to the aim of your paper, the methodology, etc. Generally speaking, several references are required to substantiate your statements, As in line 55, 76, 88, 91, 94, 100, 106, 127, 134, 160, etc. Lines 103-108 sounds like a conclusion. Line 202-211. No need of simple past. Line 302-305. I guess these lines need being deleted.

Please, reformulate the text so that it is clearer what you want to demonstrate.

Author Response

Dear Reviewer 2,

Thank you very much for valuable comments on our manuscript. We revised the manuscript carefully point-by-point, accepted and tried to meet all the recommendations of you. All the changes are clearly indicated (in yellow) in the revised manuscript.

Clarification of text concept: The text was originally written as a review. However, after submitting the manuscript, we were obliged to change the category of our manuscript to “Discussion” or “Communication” due to the shortness of the article. Hence we have chosen “Discussion”. Indeed, we wanted to present a critical review, in which we have included also our original, already published results. No experimental data has been included in the text. The information in lines 136 – 153 concerns our results published in 2006, 2007 and 2011. In this paragraph we mentioned our personal experience with the analysis of specificity of monoclonal anti-gliadin antibodies employing an ELISA test for detection of gliadin (gluten) in foodstuffs for a gluten-free diet.

On the basis of your recommendation, we reformulated the Introduction of our manuscript including clearly defined aims of the paper, methodological approaches and argumentation and information sources.

After your requirement the citations were completed and added.

We agree with the reviewer that information in lines 103 – 108 “sounds like a conclusion”. As you recommended, we incorporated the information in Conclusion (lines 292 – 297).

After your recommendation we have changed the past simple in sentences (Line 202 – 211; newly, lines: 211-220) to past perfect.

Lines 302 – 305 have been deleted.

We hope that the changes we made are acceptable for you and satisfy your requirements.

Thank you very much.

Round 2

Reviewer 1 Report

The authors have adequately addressed the comments made during the first round of review.

Author Response

Dear Reviewer 1,

Thank you very much for the statement, that we have adequately addressed the comments made during the first round of review.

Best regards

Iva Hoffmanová and Daniel Sánchez

Reviewer 2 Report

Dear Authors,

thank you for your reply and for addessing my points.

Your paper is indeed of interest for the scientific community and addresses very interesting issues about the role of oat in GFDs, and methods for its determination.

My sugegstion is however to rewrite the paper, bearing in mind it cannot be a critical review (due to the shortness), rather it is a discussion.

Based on the already available information, try to provide the discussion with a logical flow, so that the reader can easily follow your "Discussion of the matter" and see how the different points are connected.

One more suggestion for the introduction. In case you wanted to amend it only by adding your aim, move it to the end of the introduction and not at the beginning. Try to prepare the reader on how you will reach your aim, which points will be addressed in the paper.

Author Response

Dear Reviewer 2.

Thank you very much once more for your important comments and suggestion. We have accepted all of them.

1/ We are sending the manuscript as a “Discussion” type

2/ We have rewritten the manuscript according your suggestion, i.e., we have improved text of the “Introduction” (we have explained our aims of paper in details and its scientific significance), and further we have re-structuralized the body of the text. 

The new changes (changes for Round 2) are indicated by red letters. The changes introduced during the Round 1 are highlighted in yellow.

Best regards

Iva Hoffmanová and Daniel Sánchez